

# Parent Hamiltonians of Jastrow wavefunctions

**Mathieu Beau[1]⋆, Adolfo del Campo[2,3,1]**

**1** Department of Physics, University of Massachusetts, Boston, MA 02125, USA
**2** Department of Physics and Materials Science,
University of Luxembourg, L-1511 Luxembourg, Luxembourg
**3** Donostia International Physics Center, E-20018 San Sebastián, Spain

⋆ mathieu.beau.89@gmail.com

## Abstract

We find the complete family of many-body quantum Hamiltonians with ground-state of Jastrow form involving the pairwise product of a pair function in an arbitrary spatial dimension. The parent Hamiltonian generally includes a two-body pairwise potential as well as a three-body potential. We thus generalize the Calogero-Marchioro construction for the three-dimensional case to an arbitrary spatial dimension. The resulting family of models is further extended to include a one-body term representing an external potential, which gives rise to an additional long-range two-body interaction. Using this framework, we provide the generalization to an arbitrary spatial dimension of well-known systems such as the Calogero-Sutherland and Calogero-Moser models. We also introduce novel models, generalizing the McGuire many-body quantum bright soliton solution to higher dimensions and considering ground-states which involve e.g., polynomial, Gaussian, exponential, and hyperbolic pair functions. Finally, we show how the pair function can be reverse-engineered to construct models with a given potential, such as a pair-wise Yukawa potential, and to identify models governed exclusively by three-body interactions.

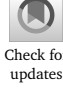

# 1 Introduction

Exactly solvable models play a prominent role in many-body physics. Their study has guided the exploration of strongly correlated states of matter, superconductivity, and other rich phenomena. It has been key to the discovery of Bose-Fermi duality and its generalizations [1] and has motivated new concepts such as the generalized exclusion statistics [2,3]. Solvable models are also utilized as test-bed for approximations and are useful in the development of nonperturbative methods [4].

Most known exact solutions are confined to one spatial dimension, in which the scattering between particles is highly constrained by conservation laws [5]. Powerful mathematical methods such as the Bethe ansatz and the quantum inverse scattering technique have been developed for their description [6–9]. The availability of solvable models in higher spatial dimensions is however scarce. A successful strategy to discover them consists of choosing the ground-state in a given form. A widely used choice is the so-called Jastrow form in which the many-body wavefunction is expressed as the pair-wise product of a two-body pair function $f$ of the interparticle distance $r_{ij}$ [10]

$$\Phi_0(\vec{r}_1, \ldots, \vec{r}_N) = \prod_{i<j} f(r_{ij}), \tag{1}$$

which captures spatial two-body correlations and has proved useful in the description of superfluid Helium and quantum fluids. The Jastrow form can be easily modified to account for external one-body potentials (such as an optical lattice or a harmonic trap) by multiplying it by a product of one-body terms. In this spirit, so-called Slater-Jastrow wave functions can be constructed as products of Jastrow functions and Slater determinants of single-particle wavefunctions, e.g., to describe electronic systems. An alternative approach to construct fermionic wave-functions starts from pair orbitals and use Pfaffian wave-functions [11,12].

In addition, generalizations of the Jastrow form to include higher-order correlations have also been proposed. One can thus consider an expansion of the form [10]

$$\Phi_0(\vec{r}_1, \ldots, \vec{r}_N) = \prod_{i<j} f(r_{ij}) \times \prod_i g(\vec{r}_i) \times \prod_{ijk} h(\vec{r}_i, \vec{r}_j, \vec{r}_k) \times \cdots \tag{2}$$

Once the ground-state wavefunction $\Phi_0$ is chosen, one can consider the explicit action of the kinetic $\hat{T}$ operator on it. Whenever it is possible to identify the terms resulting from the explicit

evaluation as an interaction potential acting on $\Phi_0$, $\hat{T}\Phi_0 = -V\Phi_0$, the parent Hamiltonian $\hat{H}_0$ of $\Phi_0$ follows, with the Schrödinger equation $\hat{H}_0\Phi_0 = (\hat{T}+V)\Phi_0 = 0$. This 'optimistic' approach to identifying exact solutions of many-body quantum systems was pioneered by Sutherland in the derivation of the Calogero-Sutherland model [13, 14]. However, and at variance with that case, the parent Hamiltonian of (1) is only expected to be quasi-exactly solvable, in the sense that only part of the spectrum may be derived. Further, the parent Hamiltonian is generally characterized by two-body and three-body interactions. The conditions under which it involves only two-body interactions have been studied under periodic boundary conditions in one spatial dimension and restrict the form of the two-body function $f$ to be a Jacobi theta function in one spatial dimension [15].

The analogous construction in one-spatial dimension without imposing periodic boundary conditions has recently been presented in [16] for Jastrow wavefunctions (2) including one and two-body functions. For many relevant choices of the pair function, the three-body term vanishes, becomes constant, or reduces to a two-body term. As a result, Jastrow ground-states are common in one-dimensional models containing only two-body interactions. Examples include the paradigmatic Lieb-Liniger model [17, 18] describing one-dimensional Bose gas with contact interactions of relevance to ultracold gases confined in tight-waveguides [19]. While in general eigenstates take the form of the Bethe ansatz, for attractive interactions the Jastrow form appears in the McGuire bright quantum soliton solution [20]. This feature is preserved upon embedding in a harmonic trap, provided the Hamiltonian is supplemented with long-range interactions [21]. In the case of hard-core repulsive interactions known as the Tonks-Girardeau gas [1], the Jastrow form is well known under harmonic confinement [22]. The latter is a specific instance of the celebrated Calogero-Sutherland model with inverse-square interactions [13, 23, 24]. This structure also appears in states of systems related by Bose-Fermi duality [1] and anyonic generalizations [25, 26].

Beyond the one-dimensional case, restricting the Jastrow form to the pair-wise product, Calogero and Marchioro [27] identified the family of parent Hamiltonians with a ground state of the form (1) in three spatial dimensions. The latter generally include two-body and three-body interactions.

In two spatial dimensions, Jastrow wavefunctions are ubiquitous in the description of quantum Hall physics with effective complex coordinates of the form $z_j = x_j - iy_j$ [28]. For example, the Laughlin state [29] can be seen as a deformation of the ground state of the one-dimensional Calogero-Sutherland model [30]. Such Jastrow wavefunctions are related to models of anyons including a relative angular momentum term [31]. For real coordinates (i.e., $\vec{r}_j = (x_j, y_j)$) and in the absence of momentum-dependent terms (other than the kinetic energy contribution), few instances of quantum many-body solvable models are available [32, 33].

In arbitrary spatial dimension, Gambardella used a group theoretical approach to identify the family of parent Hamiltonian of Jastrow ground-state wavefunctions in translationally invariant systems with $SU(1, 1)$ symmetry [34]. The latter applies to Calogero-like models with inverse-square interactions but it is rather restrictive and excludes relevant cases involving, e.g., contact and Coulomb interactions. A closely related and more general result was reported by Kane et al. [35] who considered bosonic models with translational invariance and identified the structure of the parent Hamiltonian including two and three-body terms. Further, they showed that the long-wavelength physics of these models is independent of the three-body interactions. However, the interaction terms were expressed merely in terms of gradients of the pair function, i.e., as momentum-dependent interactions. The accumulated results in different dimensions indicate that the parent Hamiltonian with ground-state of Jastrow form is generally not exactly solvable, and only part of the spectrum is available.

In this work, we provide explicitly the complete family of parent Hamiltonian in arbitrary spatial dimension $d$ with ground state of Jastrow form including one and two-body pair

functions, i.e., $\Psi_0(\vec{r}_1,\ldots,\vec{r}_N) = \prod_i g(\vec{r}_i) \prod_{i<j} f(r_{ij})$. It is shown that such models generally involve two-body and three-body interaction terms. In addition, the one-body function $g$ can be used to account for an external one-body potential such as a harmonic trap, but only when the parent Hamiltonian is supplemented with a long-range two-body contribution. Our results thus pave the way to the systematic construction of quasi-solvable models in an arbitrary spatial dimension.

## 2 Parent Hamiltonians in d-spatial dimensions

In arbitrary spatial dimension $d$, we denote the spatial coordinate of a particle with index $i$ by a vector $\vec{r}_i \in \mathbb{R}^d$ with components $r_{i,\alpha}$ $(\alpha = 1,\ldots,d)$ and norm $r_i = \|\vec{r}_i\| = \sqrt{\sum_{\alpha=1}^d r_{i,\alpha}^2}$. The kinetic energy operator is given in terms of the Laplace operator,

$$\Delta_i = \sum_{\alpha=1}^d \frac{\partial^2}{\partial r_{i,\alpha}^2}\,. \tag{3}$$

In hyperspherical coordinates for a system of $N$ particles, the explicit form of the kinetic term reads

$$\hat{T} = -\frac{\hbar^2}{2m}\sum_{i=1}^N \Delta_i = -\frac{\hbar^2}{2m}\sum_{i=1}^N \left[\frac{1}{r_i^{d-1}}\frac{\partial}{\partial r_i} r_i^{d-1}\frac{\partial}{\partial r_i} + \frac{1}{r_i^2}\Delta_i^{S^{d-1}}\right],$$

where the Laplace-Beltrami operator on the sphere $S^{d-1}$ is denoted by $\Delta_i^{S^{d-1}}$. We consider ground-states described as the pair-wise product of pair functions, that depend exclusively on the relative distance between particles $r_{ij} = \|\vec{r}_i - \vec{r}_j\|$, i.e.,

$$\Phi_0(\vec{r}_1,\ldots,\vec{r}_N) = \langle \vec{r}_1,\ldots,\vec{r}_N|\Phi_0\rangle = \prod_{i<j} f(r_{ij})\,, \tag{4}$$

which describes bosons, being symmetric with respect to permutation of particles. For this choice of $\Phi_0$, an important simplification occurs as

$$\Delta_i^{S^{d-1}}\Phi_0 = 0\,. \tag{5}$$

We are interested in finding the many-body quantum parent Hamiltonian satisfying the time-independent Schrödinger equation

$$\hat{H}_0|\Phi_n\rangle = E_n|\Phi_n\rangle\,. \tag{6}$$

In one [16] and three [27] spatial dimensions, it is known that $H_0$ involves exclusively two-body and three-body interactions

$$\hat{H}_0 = \hat{T} + V_2 + V_3\,. \tag{7}$$

We next show that the form of the parent Hamiltonian (7) holds in arbitrary spatial dimension $d$. To identify it, we explicitly compute the action of the kinetic energy operator on the Jastrow wavefunction (4). For compactness, we denote $f(r_{ij}) = f_{ij}$ and similarly for the first and second derivatives of the function $f$. As shown In Appendix A, explicit evaluation of the action of the Laplacian yields:

$$\sum_i \Delta_i \Phi_0 = \sum_i \sum_{j\neq i} \left(\frac{f_{ij}''}{f_{ij}} + \frac{d-1}{r_{ij}}\frac{f_{ij}'}{f_{ij}}\right)\Phi_0 + \sum_i \sum_{j\neq k\neq i} \frac{\vec{r}_{ij}}{r_{ij}}\cdot\frac{\vec{r}_{ik}}{r_{ik}}\frac{f_{ij}'}{f_{ij}}\frac{f_{ik}'}{f_{ik}}\Phi_0\,.$$

After noticing that the functions $f_{ij}$ and $f_{ij}''$ are symmetric with respect to permutations $\vec{r}_i \leftrightarrow \vec{r}_j$, we rewrite the first sum as $\sum_{i \neq j} = 2 \sum_{i<j}$. For the second term, we use the following sum decomposition

$$\sum_{i \neq j \neq k} A_{ijk} = 2 \sum_{i<j<k} A_{ijk} + 2 \sum_{j<k<i} A_{ijk} + 2 \sum_{k<i<j} A_{ijk} = 2 \sum_{i<j<k} \left( A_{ijk} + A_{jki} + A_{kij} \right),$$

to obtain

$$\sum_i \Delta_i \Phi_0 = 2 \sum_{i<j} \left( \frac{f_{ij}''}{f_{ij}} + \frac{d-1}{r_{ij}} \frac{f_{ij}'}{f_{ij}} \right) \Phi_0$$

$$+ 2 \sum_{i<j<k} \left( \frac{\vec{r}_{ij}}{r_{ij}} \cdot \frac{\vec{r}_{ik}}{r_{ik}} \frac{f_{ij}'}{f_{ij}} \frac{f_{ik}'}{f_{ik}} - \frac{\vec{r}_{ij}}{r_{ij}} \cdot \frac{\vec{r}_{jk}}{r_{jk}} \frac{f_{ij}'}{f_{ij}} \frac{f_{jk}'}{f_{jk}} + \frac{\vec{r}_{ik}}{r_{ik}} \cdot \frac{\vec{r}_{jk}}{r_{jk}} \frac{f_{ik}'}{f_{ik}} \frac{f_{jk}'}{f_{jk}} \right) \Phi_0 \,, \quad (8)$$

where we use $\vec{r}_{ij} = -\vec{r}_{ji}$.

As an upshot, the parent Hamiltonian of a Jastrow wavefunction in dimension $d$ takes the explicit form

$$\hat{H}_0 = \hat{T} + V_2 + V_3 \,, \tag{9}$$

$$V_2 = \frac{\hbar^2}{m} \sum_{i<j} \left[ \frac{f''(r_{ij})}{f(r_{ij})} + (d-1) \frac{f'(r_{ij})}{r_{ij} f(r_{ij})} \right], \tag{10}$$

$$V_3 = \frac{\hbar^2}{2m} \sum_i \sum_{j \neq k \neq i} \frac{\vec{r}_{ij}}{r_{ij}} \cdot \frac{\vec{r}_{ik}}{r_{ik}} \frac{f_{ij}'}{f_{ij}} \frac{f_{ik}'}{f_{ik}}$$

$$= \frac{\hbar^2}{m} \sum_{i<j<k} \left[ \frac{\vec{r}_{ij} \cdot \vec{r}_{ik}}{r_{ij} r_{ik}} \frac{f'(r_{ij}) f'(r_{ik})}{f(r_{ij}) f(r_{ik})} - \frac{\vec{r}_{ij} \cdot \vec{r}_{jk}}{r_{ij} r_{jk}} \frac{f'(r_{ij}) f'(r_{jk})}{f(r_{ij}) f(r_{jk})} + \frac{\vec{r}_{ik} \cdot \vec{r}_{jk}}{r_{ik} r_{jk}} \frac{f'(r_{ik}) f'(r_{jk})}{f(r_{ik}) f(r_{jk})} \right], \tag{11}$$

with zero ground-state energy $E_0 = 0$, i.e., $\hat{H}_0 |\Phi_n\rangle = 0$. We note the presence of a three-body term that does not vanish in general (unless $f$ is constant), as we shall see.

For $d = 3$, equations (9)-(11) reduce to the Calogero-Marchioro complete family of parent Hamiltonians in three spatial dimensions [27]. Similarly, equations (9)-(11) generalize the complete family of parent Hamiltonians in one spatial dimension identified in [16]. The $d = 1$ case is indeed better discussed as a separate instance, due to the appearance of contact interactions. In this sense, our current work focuses on $d > 1$. In what follows we proceed to the construction of instances of this family by considering relevant choices of the pair function $f(r_{ij})$, i.e., by specifying the ground-state Jastrow wavefunction. But first, we discuss how to include a one-body potential such as external confinement. To this end, we include a product over single-particle terms in the Jastrow wavefunction.

## 3 Localized Jastrow wavefunctions and confining potentials

The Jastrow form (1) is exclusively given as the pairwise product of a pair correlation function. In many applications, a one-body term is added to the Hamiltonian to account for an external potential to which all particles are subject. This is particularly relevant in the description of ultracold gases confined in a trap. In paradigmatic instances of one-dimensional integrable models such as hard-core bosons in the Tonks-Girardeau regime and the (rational) Calogero-Sutherland model, the effect of an external harmonic trap on the ground-state wavefunction is to modify the Jastrow form by including the product of a one body-term [16].

We thus consider a ground-state of the form

$$\Psi_0 = \prod_k g(r_k) \prod_{i<j} f(r_{ij}) = \prod_k g(r_k)\Phi_0 . \tag{12}$$

In spite of the fact that we know the parent Hamiltonian of $\Phi_0$, derived in the previous section, it proves convenient to perform an explicit computation making use of the Jastrow form of $\Phi_0$. The detailed calculation is shown in the Appendix A, where the Laplacian is found to be

$$\Delta_i \Psi_0 = \sum_{j\neq i} \left[ \frac{d-1}{r_{ij}} \frac{f'_{ij}}{f_{ij}} + \frac{f''_{ij}}{f_{ij}} \right] \Psi_0 + \sum_{j\neq k \neq i} \left( \frac{\vec{r}_{ij}}{r_{ij}} \cdot \frac{\vec{r}_{ik}}{r_{ik}} \frac{f'_{ij}}{f_{ij}} \frac{f'_{ik}}{f_{ik}} \right) \Psi_0$$
$$+ 2\sum_{j\neq i} \left( \frac{\vec{r}_{ij}}{r_{ij}} \frac{f'_{ij}}{f_{ij}} \cdot \frac{\vec{r}_i}{r_i} \frac{g'_i}{g_i} \right) \Psi_0 + \left[ \frac{d-1}{r_i} \frac{g'_i}{g_i} + \frac{g''_i}{g_i} \right] \Psi_0 .$$

To find the parent Hamiltonian, we evaluate the kinetic term and deduce the form of the potential $V$ using the identity

$$\hat{H}\Psi_0 = 0 , \tag{13}$$

where

$$\hat{H} = -\frac{\hbar^2}{2m} \sum_i \Delta_i + V = \hat{H}_0 + V_1 + V_{2LL} . \tag{14}$$

Using the equation above, we find that the potential $V$ includes the two-body and three-body terms $V_2$ and $V_3$ of $\hat{H}_0$, as well as an external one-body potential $V_1$, and a mixed coupling between the two-body and external potential that we denote by $V_{2LL}$ as it generally describes a long-range two body contribution:

$$V_1 = \frac{\hbar^2}{2m} \sum_i \left( \frac{d-1}{r_i} \frac{g'_i}{g_i} + \frac{g''_i}{g_i} \right) , \tag{15}$$

$$V_{2LL} = \frac{\hbar^2}{m} \sum_{i\neq j} \left( \frac{\vec{r}_{ij}}{r_{ij}} \cdot \frac{\vec{r}_i}{r_i} \frac{f'_{ij}}{f_{ij}} \frac{g'_i}{g_i} \right) = \frac{\hbar^2}{m} \sum_{i<j} \frac{f'_{ij}}{f_{ij}} \frac{\vec{r}_{ij}}{r_{ij}} \cdot \left( \frac{\vec{r}_i}{r_i} \frac{g'_i}{g_i} - \frac{\vec{r}_j}{r_j} \frac{g'_j}{g_j} \right) . \tag{16}$$

As a particular example, we consider the presence of an isotropic harmonic trap, that corresponds to the choice

$$g_i = e^{-\frac{m\omega}{2\hbar} r_i^2} . \tag{17}$$

In this case

$$V_1 = \frac{1}{2} m\omega^2 \sum_{i=1}^{N} r_i^2 - dN \frac{\hbar\omega}{2} , \tag{18}$$

which represents a harmonic trap minus the zero-point energy contribution. The coupling term reads in this case

$$V_{2LL} = -\hbar\omega \sum_{i\neq j} \left( \frac{\vec{r}_{ij}}{r_{ij}} \cdot \vec{r}_i \frac{f'_{ij}}{f_{ij}} \right) = -\hbar\omega \sum_{i<j} \frac{f'_{ij}}{f_{ij}} r_{ij} . \tag{19}$$

This term is the generalization to arbitrary spatial dimension of the two-body function long-range term found in the long-range Lieb-Liniger model [16,21]. We also note that this term reduces to a constant in the case of $SU(1,1)$ systems considered by Gambardella [34].

More generally, the role of an external spatially isotropic confining potential can be associated with the one-body function $g(r_i) = \exp[v(r_i)]$, provided that the Hamiltonian is supplemented with the $V_{2LL}$ term. Specifically, the one-body external potential reads

$$V_1 = \frac{\hbar^2}{2m} \sum_{i=1}^{N} \left[ \frac{d-1}{r} v'(r_i) + v'(r_i)^2 + v''(r_i) \right], \tag{20}$$

while the two-body long-range potential reads

$$V_{2LL} = \frac{\hbar^2}{m} \sum_{i \neq j} \left( \frac{\vec{r}_{ij}}{r_{ij}} \cdot \frac{\vec{r}_i}{r_i} \frac{f'_{ij}}{f_{ij}} v'(r_i) \right) = \frac{\hbar^2}{m} \sum_{i < j} \frac{f'_{ij}}{f_{ij}} \frac{\vec{r}_{ij}}{r_{ij}} \cdot \left( \frac{\vec{r}_i}{r_i} v'(r_j) - \frac{\vec{r}_j}{r_j} v'(r_j) \right). \tag{21}$$

These equations for $V_1$ and $V_{2LL}$ generalize the results for the embedding of Jastrow ground-states in external potentials in [16] from one to an arbitrary spatial dimension $d$.

Summarizing this section, if a wavefunction $\Phi_0(\vec{r}_1, \ldots, \vec{r}_N) = \prod_{i<j} f(r_{ij})$ fulfills the Schrödinger equation $(\hat{T} + V_2 + V_3)\Phi_0 = 0$, then the modified wavefunction $\Psi_0 = \prod_i e^{v(r_i)} \Phi_0$ obeys the Schrödinger equation

$$\hat{H}\Psi_0 = (\hat{T} + V_1 + V_2 + V_{2LL} + V_3)\Psi_0 = 0, \tag{22}$$

with $V_1$ and $V_{2LL}$ given by Eqs. (20) and (21), respectively.

# 4 List of models

The family of parent Hamiltonians of Jastrow wavefunction is infinite. To determine specific instances within this family it suffices to specify a valid pair function $f$. We next discuss some specific examples, partially motivated by the existence of analogous models in one spatial dimension:

- Calogero-Moser (CM) model: $f_{ij} = r_{ij}^{\lambda}$.

- Calogero-Sutherland (CS) model: $f_{ij} = r_{ij}^{\lambda} e^{-\frac{\omega}{2} r_{ij}^2}$.

- McGuire model: $f_{ij} = e^{-c r_{ij}}$, $c > 0$.

- Hyperbolic (inverse-sinh-square) model: $f_{ij} = \sinh(r_{ij}/r_0)^{\lambda}$, $\lambda > 0$.

- New model 1: McGuire-Calogero-Sutherland: $f_{ij} = e^{c r_{ij}} e^{-\frac{\omega}{2} r_{ij}^2}$.

- New model 2: McGuire-Calogero-Moser model: $f_{ij} = r_{ij}^{\lambda} e^{-c r_{ij}}$, $c > 0$.

- New model 3: Hyperbolic McGuire model: $f_{ij} = \sinh(r_{ij}/r_0)^{\lambda} e^{-c r_{ij}}$, $c > 0$.

- New model 5: Hyperbolic Calogero-Sutherland model: $f_{ij} = \sinh(r_{ij}/r_0)^{\lambda} e^{-\frac{\omega}{2} r_{ij}^2}$.

- New model 6: Model with Yukawa-like pairwise interactions: $f_{ij} = r_{ij}^{\lambda} e^{a r_{ij} + b r_{ij}^2 + c r_{ij}^3}$.

## 4.1 Calogero-Moser model in d-spatial dimensions

In one spatial dimension, the pair function

$$f(r_{ij}) = r_{ij}^{\lambda}, \tag{23}$$

for the Jastrow wavefunction is associated with the celebrated Calogero-Moser model as parent Hamiltonian. For this choice $V_3 = 0$, and the CS Hamiltonian exclusively involves two-body interactions that decay with the square of the interparticle distance.

The $d$-dimensional case, obtained from Eqs. (10)-(11), and described by the Hamiltonian

$$\hat{H}_0 = -\frac{\hbar^2}{2m} \sum_{i=1}^{N} \Delta_i + \frac{\hbar^2}{m} \sum_{i<j} \frac{\lambda(\lambda + d - 2)}{|r_{ij}|^2} + V_3, \tag{24}$$

with

$$V_3 = \frac{\hbar^2 \lambda^2}{m} \sum_{i<j<k} \left[ \frac{\vec{r}_{ij} \cdot \vec{r}_{ik}}{r_{ij}^2 r_{ik}^2} - \frac{\vec{r}_{ij} \cdot \vec{r}_{jk}}{r_{ij}^2 r_{jk}^2} + \frac{\vec{r}_{ik} \cdot \vec{r}_{jk}}{r_{ik}^2 r_{jk}^2} \right]. \tag{25}$$

In the $d = 1$ case, the latter reduces to a constant term. In arbitrary $d$, the Hamiltonian preserves $SU(1,1)$ symmetry. Embedding in a harmonic trap results in an additional long-range pairwise interaction term (Eq. (21) that in this case takes becomes a constant

$$V_{\text{2LL}} = -\frac{\hbar \omega \lambda}{2} N(N-1). \tag{26}$$

The resulting Hamiltonian was discussed by Khare and Ray in [32, 33], who also provided a tower of excited states. We note that the interaction terms of the Hamiltonian have the same scaling dimension as the kinetic energy operator. Under variations of the trap-frequency $\omega \to \omega(t)$, the time-evolution is thus self-similar. Exact coherent states can thus be constructed following [14, 36, 37]. In addition, the homogeneous character of $f(r_{ij})$ makes it possible to study a wide range of properties including the mean energy [38] and energy fluctuations [39], as well as information-theoretic quantities such as the time-dependent fidelity and Bures angle [40].

## 4.2 Calogero-Sutherland model d-spatial dimensions

Consider the two-body function of the Calogero-Sutherland (CS) model

$$f_{ij} = r_{ij}^{\lambda} e^{-\frac{\mu \Omega}{2\hbar} r_{ij}^2}. \tag{27}$$

The corresponding two-body term involves harmonic and inverse-square interactions

$$V_2 = -\frac{\hbar \mu \Omega N(N-1)}{2m}(2\lambda + d) + \sum_{i<j} \left( \frac{\mu^2}{m} \Omega^2 r_{ij}^2 + \frac{\hbar^2}{m} \frac{\lambda(\lambda + d + 2)}{r_{ij}^2} \right). \tag{28}$$

The three-body term, written in compact form, reads

$$V_3 = \frac{\hbar^2}{2m} \sum_i \sum_{j \neq k \neq i} \vec{r}_{ij} \cdot \vec{r}_{ik} \left[ \frac{\mu^2 \Omega^2}{\hbar^2} + \frac{\lambda^2}{r_{ij}^2 r_{ik}^2} - \frac{\mu \Omega}{\hbar} \left( \frac{1}{r_{ij}^2} + \frac{1}{r_{ik}^2} \right) \right]. \tag{29}$$

In this case, embedding in a harmonic trap of frequency $\omega$ results in an additional harmonic contribution

$$V_{\text{2LL}} = \mu \omega \Omega \sum_{i<j} r_{ij}^2 - \frac{\hbar \omega \lambda}{2} N(N-1). \tag{30}$$

### 4.3 Bose gas with contact and Coulomb-like inverse-distance interactions in d-spatial dimensions

In $d = 1$, the attractive one-dimensional Bose gas with contact interactions, known as the Lieb-Liniger model [17, 18], supports quantum bright soliton states described by the McGuire wavefunction $\Phi_0 = e^{-c\sum_{i<j}|x_{ij}|}$ [20]. We next consider the generalization to $d > 1$, in which $\Phi_0$ is determined by the pair function

$$f_{ij} = e^{-cr_{ij}}, \; c > 0 \,. \tag{31}$$

Explicit computation yields

$$V_2 = \frac{\hbar^2 N(N-1)}{2m}c^2 - (d-1)c\sum_{i<j}\frac{1}{r_{ij}}\,. \tag{32}$$

Note that $V_2$ takes the form of a gravitational or Coulomb-like potential in $d = 3$. However, we recall that in $d = 2$ the latter involves a logarithmic dependence on the relative coordinate, rather than an inverse power-law. In the $d = 1$ case, the Coulomb and gravitational potentials are linear on the relative distance between particles. As a result, for $d \neq 3$, the power-law interaction $\sim 1/r_{ij}$ does not admit an analogy with electromagnetism or Newtonian gravity.

Regarding the three-body contribution, it takes a particularly simple form given by

$$V_3 = \frac{\hbar^2 c^2}{m}\sum_{i<j<k}\left[\frac{\vec{r}_{ij}\cdot\vec{r}_{ik}}{r_{ij}r_{ik}} - \frac{\vec{r}_{ij}\cdot\vec{r}_{jk}}{r_{ij}r_{jk}} + \frac{\vec{r}_{ik}\cdot\vec{r}_{jk}}{r_{ik}r_{jk}}\right] = \frac{\hbar^2 c^2}{m}\sum_{i<j<k}\left(\cos(\theta_{i,jk}) + \cos(\theta_{j,ki}) + \cos(\theta_{k,ij})\right),$$

where $\theta_{i,jk} = \frac{\vec{r}_{ij}\cdot\vec{r}_{ik}}{r_{ij}r_{ik}}$ is the angle between the relative positions $\vec{r}_{ij}$ and $\vec{r}_{ik}$. Interestingly, the sum of the cosines varies between 1 and 3/2 depending on the relative positions of three particle, e.g., it takes unit value if the three particles are aligned and equals 3/2 if they form an equilateral triangles. This observation brings us to find the lower and the upper bound of the three-body potential

$$\frac{\hbar^2 c^2}{m}\frac{N(N-1)(N-2)}{6} \leq V_3 \leq \frac{\hbar^2 c^2}{m}\frac{N(N-1)(N-2)}{4} \,, \tag{33}$$

which is consistent with [34] (equation (27) and comment before that). Notice that for $d = 1$, we find that the three-body term is constant and is equal to the lower bound above [21]. From the observation above, the ground-state energy is minimized in a classical configuration in which particles are located at the apex of $d$-dimensional regular simplex blocks (e.g., equilateral triangles for $d = 2$, tetrahedron for $d = 3$) with edges of characteristic length $a = 1/c$.

The embedding of such state in an isotropic harmonic trap (18) is characterized by the wavefunction

$$\Psi_0 = e^{-c\sum_{i<j}r_{ij}}e^{-\frac{m\omega}{2\hbar}\sum_i r_i^2} \,, \tag{34}$$

whenever the Hamiltonian is supplemented by the long-range two-body term

$$V_{2LL} = \hbar\omega c\sum_{i\neq j}\left(\frac{\vec{r}_{ij}}{r_{ij}}\cdot\vec{r}_i\right) = \hbar\omega c\sum_{i<j}r_{ij} \tag{35}$$

and by the external potential (18). This can be seen as a higher-dimensional generalization of the confinement-induced long-range term in the modified Lieb-Liniger model [16, 21].

### 4.4 Inverse-sinh-square potentials in d-spatial dimensions

In $d = 1$, the pair correlation function $\sinh(|x_i - x_j|)$ constitutes a relevant example and is associated with a parent Hamiltonian characterized by an inverse-sinh-square pairwise potential, often referred to as a hyberbolic potential for short [15, 16]. It is natural to consider its higher dimensional generalization associated with the pair function

$$f_{ij} = \sinh(r_{ij}/r_0)^\lambda, \tag{36}$$

where $r_0$ as units of length. This choice imposes a hard-core constraint on $\Phi_0$ which vanishes as $r_{ij} \to 0$. Further, at long distances the pair function behaves as an exponential function $f_{ij} \sim \exp(\lambda r_{ij}/r_0)$ over the range $r_0/\lambda$. In this case, the two-body term reads

$$V_2 = \frac{\hbar^2 \lambda^2 N(N-1)}{2mr_0^2} + \frac{\hbar^2}{m} \sum_{i<j} \left( \frac{\lambda(\lambda-1)}{r_0^2 \sinh(r_{ij}/r_0)^2} + \frac{\lambda(d-1)}{r_0} \frac{1}{r_{ij}} \coth(r_{ij}/r_0) \right). \tag{37}$$

Interestingly, at short distances, this potential behaves as

$$V_2 = \frac{\hbar^2 \lambda(2\lambda+d)N(N-1)}{6mr_0^2} + \frac{\hbar^2}{m} \sum_{i<j} \left( \frac{\lambda(\lambda+d-2)}{r_{ij}^2} + \frac{\lambda(3\lambda-d-2)}{45r_0^4} r_{ij}^2 \right), \tag{38}$$

which effectively takes the form of that in the Calogero-Sutherland model.
By contrast for $r_{ij}/r_0 \gg 1$,

$$V_2 = \frac{\hbar^2 \lambda^2 N(N-1)}{2mr_0^2} + \frac{\hbar^2}{m} \sum_{i<j} \frac{\lambda(d-1)}{r_0 r_{ij}}, \tag{39}$$

which takes the form of the Coulomb-like inverse-distance interaction.
    In addition, the three-body contribution reads

$$V_3 = \frac{\hbar^2 \lambda^2}{2mr_0^2} \sum_i \sum_{j \neq k \neq i} \vec{r}_{ij} \cdot \vec{r}_{ik} \frac{\coth(r_{ij}/r_0)\coth(r_{ij}/r_0)}{r_{ij}r_{ik}}. \tag{40}$$

    Regarding the embedding in a harmonic trap of frequency $\omega$, it gives rise to the additional interaction term

$$V_{2LL} = -\frac{\hbar\omega\lambda}{r_0} \sum_{i<j} r_{ij} \coth(r_{ij}/r_0), \tag{41}$$

which is continuous and effectively harmonic near the origin, as in the $d = 1$ case [16], given that $(r/r_0)\coth(r/r_0) \approx 1 + (r/r_0)^2/3 + \mathcal{O}(r/r_0)^2$.

### 4.5 McGuire-Calogero-Sutherland model (MCS)

Consider the pair correlation function

$$f_{ij} = e^{cr_{ij}} e^{-\mu\Omega r_{ij}^2/(2\hbar)}, \tag{42}$$

with first and second spatial derivatives given by

$$f'_{ij} = cf - \frac{\mu\Omega}{\hbar} r_{ij} f_{ij} = \left(c - \frac{\mu\Omega}{\hbar} r_{ij}\right) f_{ij}, \tag{43}$$

$$f''_{ij} = -\frac{\mu\Omega}{\hbar} f_{ij} + \left(c - \frac{\mu\Omega}{\hbar} r_{ij}\right)^2 f_{ij} = \left(-\frac{\mu\Omega}{\hbar} + c - \frac{2\mu\Omega c}{\hbar} r_{ij} + \left(\frac{\mu\Omega}{\hbar}\right)^2 r_{ij}^2\right) f_{ij}. \tag{44}$$

We note the following identities

$$\frac{d-1}{r_{ij}}\frac{f'_{ij}}{f_{ij}} = c\frac{d-1}{r_{ij}} - \frac{\mu\Omega}{\hbar}(d-1), \tag{45}$$

$$\frac{f'_{ij}}{f_{ij}}\frac{f'_{ik}}{f_{ik}} = (c-\mu\Omega r_{ij})(c-\frac{\mu\Omega}{\hbar}r_{ik}) = c^2 + \left(\frac{\mu\Omega}{\hbar}\right)^2 r_{ij}r_{ik} - \frac{\mu\Omega c}{\hbar}(r_{ij}+r_{ik}), \tag{46}$$

$$\frac{\vec{r}_{ij}\cdot\vec{r}_{ik}}{r_{ij}r_{ik}}\frac{f'_{ij}}{f_{ij}}\frac{f'_{ik}}{f_{ik}} = c^2\frac{\vec{r}_{ij}\cdot\vec{r}_{ik}}{r_{ij}r_{ik}} + \left(\frac{\mu\Omega}{\hbar}\right)^2\vec{r}_{ij}\cdot\vec{r}_{ik} - \frac{\mu\Omega c}{\hbar}\left(\vec{r}_{ij}\cdot\frac{\vec{r}_{ik}}{r_{ik}} + \vec{r}_{ik}\cdot\frac{\vec{r}_{ij}}{r_{ij}}\right). \tag{47}$$

Using the first one in combination with equation (10), we find

$$V_2 = V_2^{(1)} + V_2^{(2)},$$

where

$$V_2^{(1)} = \frac{\hbar^2}{2m}N(N-1)\left(c-\frac{\mu\Omega}{\hbar}\right) - 2\gamma\hbar\Omega c\sum_{i<j}r_{ij} + \gamma\mu\Omega^2\sum_{i<j}r_{ij}^2, \tag{48}$$

$$V_2^{(2)} = \frac{\hbar^2 c}{m}(d-1)\sum_{i<j}\frac{1}{r_{ij}} - \frac{\hbar\Omega}{2}\gamma(d-1)N(N-1), \tag{49}$$

where $\gamma = \mu/m$. As for the three-body term, using equations (46) and (47) together with (11), we find

$$V_3 = V_3^{(1)} + V_3^{(1)} + V_3^{(1)}, \tag{50}$$

where

$$V_3^{(1)} = \frac{\hbar^2 c^2}{m}\sum_{i<j<k}\left(\frac{\vec{r}_{ij}\cdot\vec{r}_{ik}}{r_{ij}r_{ik}} - \frac{\vec{r}_{ij}\cdot\vec{r}_{jk}}{r_{ij}r_{jk}} + \frac{\vec{r}_{ik}\cdot\vec{r}_{jk}}{r_{ik}r_{jk}}\right), \tag{51}$$

$$V_3^{(2)} = \gamma\mu\Omega^2\sum_{i<j<k}\left(\vec{r}_{ij}\cdot\vec{r}_{ik} - \vec{r}_{ij}\cdot\vec{r}_{jk} + \vec{r}_{ik}\cdot\vec{r}_{jk}\right), \tag{52}$$

$$V_3^{(3)} = -\gamma\hbar\Omega c\sum_{i<j<k}\left(\vec{r}_{ij}\cdot\vec{r}_{ik}\left(\frac{1}{r_{ik}}+\frac{1}{r_{ij}}\right) - \vec{r}_{ij}\cdot\vec{r}_{jk}\left(\frac{1}{r_{ij}}+\frac{1}{r_{jk}}\right)\right.$$
$$\left. + \vec{r}_{ik}\cdot\vec{r}_{jk}\left(\frac{1}{r_{ik}}+\frac{1}{r_{jk}}\right)\right). \tag{53}$$

As we have seen above, the first three-body term reduces to

$$V_3^{(1)} = \frac{\hbar^2}{m}(c^2+\delta^2)\frac{N(N-1)(N-2)}{6}, \tag{54}$$

where we emphasize that $\delta$ is coordinate-dependent $0 \le \delta^2 \le \frac{c^2}{2}$. The other three-body terms reduce to two-body interactions of the form

$$V_3^{(2)} = \gamma\frac{\mu\Omega^2}{2}\sum_{i<j<k}\left(r_{ik}^2+r_{ij}^2+r_{jk}^2\right) = \gamma\frac{(N-2)\mu\Omega^2}{2}\sum_{i<j}r_{ij}^2 \tag{55}$$

and

$$V_3^{(3)} = -\gamma\hbar\Omega c\sum_{i<j<k}\left(r_{ik}+r_{ij}+r_{jk}\right) = -\gamma\hbar\Omega c(N-2)\sum_{i<j}r_{ij}. \tag{56}$$

Now, combining the equations above, we find

$$\left(\frac{\hbar^2}{2m}\Delta + V_{2MCS} + V_{3MCS}\right)\phi_0 = E_0\phi_0, \tag{57}$$

with the effective two-body potential

$$V_{2MCS} = -\gamma\hbar\Omega cN\sum_{i<j} r_{ij} + \frac{\hbar^2 c}{m}(d-1)\sum_{i<j}\frac{1}{r_{ij}} + \gamma\frac{\mu\Omega^2}{2}N\sum_{i<j}r_{ij}^2 \tag{58}$$

and the three-body interactions

$$V_{3MCS} = \frac{\hbar^2}{m}(c^2 + \delta^2)\frac{N(N-1)(N-2)}{6}, \tag{59}$$

where the coordinate-dependent potential term fulfills $0 \le \delta^2 \le \frac{c^2}{2}$, and the effective zero-point energy reads

$$E_0 = \gamma\frac{\hbar\Omega d}{2}N(N-1) - \frac{\hbar^2 c}{2m}N(N-1). \tag{60}$$

Interestingly, the model is equivalent to the model described in section 4.3 for $\gamma = 0$ or $\Omega = 0$ and to the latter model in an external harmonic trap (with the additional two-body interaction (35)) for $\gamma = 1$, $\mu = m$, and $\Omega = \omega_0/N$. To see that, one can use the identity

$$\sum_i r_i^2 = NR^2 + \frac{1}{N}\sum_{i<j}r_{ij}^2, \quad \vec{R} = \frac{1}{N}\sum_i\vec{r}_i \tag{61}$$

and multiply the wavefunction (42) by the independent center of mass contribution $e^{-N(m\omega/\hbar)R^2}$, which cancels out and gives the wavefunction (34).

## 4.6 McGuire-Calogero-Moser model in d-spatial dimensions

The preceding examples provide $d$-dimensional generalizations of well-known models. The potential of our framework to guide the discovery of new quasi-exactly solvable models is apparent from the following example. Consider a two-body function

$$f_{ij} = r_{ij}^\lambda e^{-cr_{ij}}, \quad c \ge 0, \lambda > 0, \tag{62}$$

which yields to Jastrow ground-state wavefunctions which is the product of the McGuire solution of the attractive Lieb-Liniger model and that in the Calogero-Moser model. In this case

$$V_2 = \frac{\hbar^2 cN(N-1)}{2m} + \frac{\hbar^2}{m}\sum_{i<j}\left(-\frac{c(2\lambda+d-1)}{r_{ij}} + \frac{\lambda(\lambda+d-2)}{r_{ij}^2}\right), \tag{63}$$

which includes a inverse-distance interaction term (matching the Coulomb/gravitational one in $d = 3$) together with an inverse-square interaction. This combination is reminiscent of the Kratzer's molecular potential [41].

Given the fact that $f'/f = -c + \lambda/r$, the three-body term admits the form

$$V_3 = \frac{\hbar^2 c^2}{6m}N(N-1)(N-2) + \frac{\hbar^2}{2m}\sum_i\sum_{j\neq k\neq i}\vec{r}_{ij}\cdot\vec{r}_{ik}\left[-2c\lambda\left(\frac{1}{r_{ij}^2 r_{ik}} + \frac{1}{r_{ij}r_{ik}^2}\right) + \frac{\lambda^2}{r_{ij}^2 r_{ik}^2}\right]. \tag{64}$$

The long-range two-body term stemming from the embedding in a harmonic trap of frequency $\omega$ takes the form

$$V_{2\text{LL}} = \hbar\omega c \sum_{i<j} r_{ij} - \frac{\hbar\omega\lambda}{2} N(N-1), \tag{65}$$

which is precisely the sum of the corresponding $V_{2\text{LL}}$ in Eq. (26) and Eq. (35).

The one-dimensional case seems not to have been discussed in the literature and merits some specific attention as the three-body contribution identically vanish. In particular, one finds

$$\hat{H}_0 = -\frac{\hbar^2}{2m} \sum_{i=1}^{N} \Delta_i + \frac{\hbar^2}{m} \sum_{i<j} \left( \frac{2c\lambda}{r_{ij}} + \frac{\lambda(\lambda-1)}{r_{ij}^2} \right) + \frac{\hbar^2 c^2}{2m}(N^2-1)N, \tag{66}$$

with the last term accounting for the ground-state energy of the McGuire quantum soliton. We note however that the inverse square interactions involve a hard-core constraint and thus the case $\lambda = 0$ is to be treated independently as in Sec. 4.3.

## 4.7 Hyperbolic McGuire model in d-spatial dimensions

Consider a two-body function

$$f_{ij} = \sinh(r_{ij}/r_0)^\lambda e^{-cr_{ij}}, \ c > 0, \tag{67}$$

which is the product of the pair functions in the McGuire solution and the hyperbolic model. We identify the two-body interaction term

$$\begin{aligned} V_2 = {}& \frac{\hbar^2 N(N-1)}{2m} \left( c^2 + \frac{\lambda}{r_0^2} \right) \\ & + \frac{\hbar^2}{m} \sum_{i<j} \left( \frac{\lambda(d-1)}{r_0 r_{ij}} \coth(r_{ij}/r_0) + \frac{\lambda(\lambda-1)}{r_0^2} \coth(r_{ij}/r_0)^2 - \frac{c(d-1)}{r_{ij}} + \frac{2c\lambda}{r_0} \coth(r_{ij}/r_0) \right), \end{aligned} \tag{68}$$

while the three-body term reads

$$\begin{aligned} V_3 = {}& \frac{\hbar^2}{2m} \sum_i \sum_{j \neq k \neq i} \frac{\vec{r}_{ij} \cdot \vec{r}_{ik}}{r_{ij} r_{ik}} \Bigg[ c^2 + \frac{\lambda^2}{r_0^2} \coth(r_{ij}/r_0) \coth(r_{ik}/r_0) \\ & \qquad\qquad\qquad - \frac{c\lambda}{r_0} \big( \coth(r_{ij}/r_0) + \coth(r_{ik}/r_0) \big) \Bigg]. \end{aligned} \tag{69}$$

## 4.8 Hyperbolic Calogero-Sutherland model in d-spatial dimensions

For completeness, we consider the modification of the predecing model in which the exponential decay of the pair function $f_{ij}$ is replaced by a Gaussian function. Consider a two-body function

$$f_{ij} = \sinh(r_{ij}/r_0)^\lambda e^{-\frac{\mu\Omega}{2\hbar} r_{ij}^2}, \ c > 0. \tag{70}$$

The two-body interaction term has multiple contributions

$$\begin{aligned} V_2 = {}& \frac{\hbar^2 N(N-1)}{2m} \left( -\frac{d\mu\Omega}{\hbar} + \frac{\lambda}{r_0^2} \right) \\ & + \frac{\hbar^2}{m} \sum_{i<j} \left( \frac{\lambda(d-1)}{r_0 r_{ij}} \coth(r_{ij}/r_0) + \frac{\lambda(\lambda-1)}{r_0^2} \coth(r_{ij}/r_0)^2 + \frac{\mu^2\Omega^2}{\hbar^2} r_{ij}^2 - \frac{2\mu\Omega\lambda}{\hbar r_0} r_{ij} \coth(r_{ij}/r_0) \right), \end{aligned} \tag{71}$$

which differs from that in the preceding model in the first contribution to the zero-point energy and the last two terms proportional to $r_{ij}$. Likewise, given the identity $f(r)' = -\mu\Omega r/\hbar + \lambda\coth(r/r_0)/r_0$, the three-body potential reads

$$V_3 = \frac{\hbar^2}{2m}\sum_i\sum_{j\neq k\neq i}\frac{\vec{r}_{ij}\cdot\vec{r}_{ik}}{r_{ij}r_{ik}}\left[\frac{\mu^2\Omega^2}{\hbar^2}r_{ij}r_{ik} + \frac{\lambda^2}{r_0^2}\coth(r_{ij}/r_0)\coth(r_{ik}/r_0)\right.$$
$$\left. -\frac{\mu\Omega\lambda}{r_0}\Big(r_{ik}\coth(r_{ij}/r_0) + r_{ij}\coth(r_{ik}/r_0)\Big)\right]. \quad (72)$$

### 4.9 Model with Yukawa-like pairwise interactions

The Yukawa potential has the form [42]

$$V_{\text{Yuk}}(r) = -\alpha\frac{e^{-r/D}}{r} = -V_0\frac{e^{-\delta\rho}}{\rho}\ , \quad (73)$$

where $D$ and $\alpha$ are two constants and $r$ is the relative radius between two particles, $a_0 = \hbar^2/(m\alpha)$ is the Bohr radius, $\delta = a_0/D$ is a dimensionless parameter, $\rho = r/a_0$, and $V_0 = \frac{\hbar^2}{ma_0^2}$ is the amplitude of energy of the potential. In most physical systems where the Yukawa potential is introduced, one considers the constant $D$ to be large compared to the Bohr radius, i.e., $\delta \ll 1$. Then, the Yukawa potential can be approximate as

$$V_{\text{Yuk}}(r) \approx -V_0\left(\frac{1}{\rho} + \delta - \frac{\delta^2}{2}\rho + \frac{\delta^3}{6}\rho^2 + O(\delta^4)\right)\ , \quad (74)$$

where we neglect the terms of order higher than four.

In this section, we propose to use our technique to find an approximation of the ground state of the Hamiltonian

$$H = \frac{\hbar^2}{2m}\Delta_{\vec{r}} + V_{\text{Yuk}}(r) \approx V_0\left[-\frac{\Delta_{\vec{\rho}}}{2} - \left(\frac{1}{\rho} + \delta - \frac{\delta^2}{2}\rho + \frac{\delta^3}{6}\rho^2\right)\right]\ , \quad (75)$$

where we rescaled the relative position $\vec{\rho} = \vec{r}/a_0$. Furthermore, we propose to generalize to $N$ particles with the following pairwise function

$$f_{ij} = e^{a\rho_{ij}+b\rho_{ij}^2+c\rho_{ij}^3}\ , \quad (76)$$

where $a, b, c$ are three real constant and where $\rho_{ij}$ is the dimensionless relative distance between two particles with indices $i$ and $j$, respectively. Using the identities

$$f'_{ij} = \left(a + 2b\rho_{ij} + 3c\rho_{ij}^2\right)f_{ij}\ , \quad (77a)$$

$$f''_{ij} = \left(a + 2b\rho_{ij} + 3c\rho_{ij}^2\right)^2 f_{ij} + \left(2b + 6\rho_{ij}\right)f_{ij}\ , \quad (77b)$$

we find

$$\frac{2f'_{ij}}{\rho_{ij}f_{ij}} = \frac{2a}{\rho_{ij}} + 4b + 6c\rho_{ij}\ , \quad (78a)$$

$$\frac{f''_{ij}}{f_{ij}} = (a^2 + 2b) + (4ab + 6c)\rho_{ij} + (4b^2 + 6ac)\rho_{ij}^2 + 12bc\rho_{ij}^3 + 9c^2\rho_{ij}^4\ , \quad (78b)$$

whence it follows that the two-body rescaled potential $v_2 = V_2/V_0$ equals

$$v_2(\rho_{ij}) = \left[ \frac{a}{\rho_{ij}} + \frac{1}{2}(a^2 + 6b) + (2ab + 6c)\rho_{ij} + (2b^2 + 3ac)\rho_{ij}^2 + 6bc\rho_{ij}^3 + \frac{9}{2}c^2\rho_{ij}^4 \right] . \quad (79)$$

As we did in the previous sections, the three-body potential $V_3$ can be obtained from equation (11) in a similar fashion.

Let us now take $N = 2$ and find the coefficients $a, b, c$. After identifying the coefficients in equations (75) and (79), we obtain

$$\begin{cases} a = -1 \\ b = \frac{1}{4}\left(1 - \sqrt{1 + \frac{4}{3}\delta^3 - 2\delta^2}\right) \approx \frac{1}{4}\delta^2 - \frac{1}{6}\delta^3 \\ c = -\frac{1}{2}\delta^2 + 2b \approx -\frac{1}{18}\delta^3 \end{cases} , \quad (80)$$

which leads to the potential given by equation (74) and to the ground-state energy

$$E = V_0\epsilon_0 , \quad \epsilon_0 = -\frac{1}{2}a^2 - 3b = -\frac{1}{2} - \frac{3}{4}\delta^2 + \frac{1}{2}\delta^3 . \quad (81)$$

This is consistent with results recently reported in [43], where the authors used the quantum supersymmetry approach. The advantage of our present method is that it works for any dimensions $d \geq 1$ and that it can easily extended to higher order of $\delta$ as well as to non-zero angular momentum $l > 0$. Indeed, to incorporate the angular momentum, it suffices to multiply the pairwise function (76) by $r_{ij}^l$

$$f_{ij} = r_{ij}^l e^{a\rho_{ij} + b\rho_{ij}^2 + c\rho_{ij}^3} . \quad (82)$$

We then find an additional effective potential $V_l = V_0 l(l+1)/r_{ij}^2$ and modified two-body potentials. Using similar method, we identify the constants to find

$$\begin{cases} a = -\frac{1}{1+l} \\ b \approx \frac{1+l}{4}\delta^2 - \frac{(2+l)(1+l)^2}{12}\delta^3 \\ c \approx -\frac{1+l}{18}\delta^3 , \end{cases} \quad (83)$$

and the energy level $E_l = V_0\epsilon_l$ with

$$\epsilon_l = -\frac{1}{2}a^2 - 3b - 2bl = -\frac{1}{2(1+l)^2} - \frac{3}{4}(1+l)\left(1 + \frac{2}{3}l\right)\delta^2 + \frac{1}{4}(2+l)(1+l)^2\left(1 + \frac{2}{3}l\right)\delta^3 \quad (84)$$

$$= -\frac{1}{2n^2} - \frac{1}{4}n(2n+1)\delta^2 + \frac{1}{12}(n+1)n^2(2n+1)\delta^3 , \quad (85)$$

where the quantum number $n = 1 + l$. Notice that for $\delta = 0$, we find that $E_n = \frac{E_0}{n^2}$ where $E_0 = -V_0/2 = -\hbar^2/(2ma_0^2)$ as expected. Notice that using our technique we find the same energy levels and wavefunction as in [43]. It is also possible to find the approximate solution for the higher order terms in $\delta$. The general method consists of adding power of $\rho$ in the exponential in equation (82):

$$f_{ij} = r_{ij}^l e^{\sum_{k=1}^{\infty} a_k \rho^k} \quad (86)$$

and to identify the coefficients in front of the two-body potential. One can use analytical or numerical methods to find the coefficients $a_k$, $k = 1, 2, 3, \ldots$ up to a certain order $M > 3$.

Once we identify the coefficients, we can easily find the expression of the energy levels $E_{n,l}$ for $n = l + 1$. To find the eigenstates for other degeneracies (such as $n = 1 + p + l$, $p = 1, 2, \ldots$), we have to multiply the pairwise functions (82) (for $M = 3$) or (86) (for $M > 3$) by some polynomials $\sum_{j=1}^{s} c_j r^j$ and find for which values of the coefficients $c_j$ the function satisfies the Schrödinger equation. In the limit $\delta \to 0$, these polynomial should approach the Laguerre polynomials [44]. This detail analysis is beyond the scope of this paper and would require further investigation. We note that this technique could be also used to find solutions of Schrödinger equations with potential written as a Taylor series $V(r) = \sum_{j=0}^{\infty} b_j r^j$.

## 5  Reverse-engineering pair function for given interactions

The models discussed have been derived making a choice of the pair function that singles out a given Jastrow wavefunction. Such choice can be motivated on physical grounds, by analogy with other models, etc. In other applications, one may be interested in studying models with a given kind of interaction. It is then possible to reverse engineer the form of the pair function $f_{ij}$. Indeed, by looking at the general expression of the two-body potential (10), we consider the differential equation

$$\left[ \frac{f''(r_{ij})}{f(r_{ij})} + (d-1) \frac{f'(r_{ij})}{r_{ij} f(r_{ij})} \right] = \frac{1}{r_0^2} v(r_{ij}/r_0), \tag{87}$$

where $v(r_{ij}/r_0)$ is a dimensionless potential function. Such ordinary second-order differential equation can be integrated numerically. In some cases, it admits an analytical solution.

For the sake of illustration let us consider models with vanishing two-body potential. As an interesting precedent in the literature, we note that systems of bosons dominated by three-body hard-core interactions have been introduced by Paredes *et al.* [45] in the quest of non-Abelian anyons in one dimension. The latter were further discussed in Girardeau's last solo paper [46].

In what follows we consider parent Hamiltonians of Jastrow wavefunctions in $d$ spatial dimensions with vanishing $V_2$ and governed by $V_3$. Let us first look into the case of $N = 2$ particles in $d = 3$, in which there are no interactions, i.e., the particles are free. According to the symmetry with respect to the center of mass, the solution looks like $\frac{A}{r} e^{-cr}$, where $c = \sqrt{2mE/\hbar^2}$. This is nothing but the solution of the free Schrödinger equation using spherical symmetry. It motivates the choice of the pairwise function

$$f_{ij} = \frac{A}{r_{ij}} e^{-cr_{ij}}. \tag{88}$$

Interestingly, this is an specific instance of the case discussed in section 4.6, see equation (62) with $\lambda = -1$. Indeed, plugging $\lambda = -1$ into equation (63), we find that $V_2 = 0$, which is consistent with the reasoning above. In this case, the three-body potential is given by equation (64) (again for $\lambda = -1$) and the total three-dimensional Hamiltonian reads

$$\hat{H}_0 = -\frac{\hbar^2}{2m} \sum_{i=1}^{N} \Delta_i + \frac{\hbar^2 c^2}{6m} N(N-1)(N-2)$$
$$+ \frac{\hbar^2}{2m} \sum_i \sum_{j \neq k \neq i} \vec{r}_{ij} \cdot \vec{r}_{ik} \left[ 2c \left( \frac{1}{r_{ij}^2 r_{ik}} + \frac{1}{r_{ij} r_{ik}^2} \right) + \frac{1}{r_{ij}^2 r_{ik}^2} \right]. \tag{89}$$

The family of models introduced is infinity and we conclude here our investigation of quasi-exactly solvable many-body quantum models in spatial dimension $d$. Many other models can

be found, such as those with pair-wise function given in terms of products of elementary functions (e.g. $f_{ij} = r_{ij}^{\lambda} e^{-c r_{ij}} e^{-\frac{\mu \Omega}{2\hbar} r_{ij}^2}$), considering other elementary functions (e.g. $e^{-c r_{ij}^{\alpha}}$), etc. The identification of these models may be assisted by making use of methods in supersymmetric quantum mechanics [47].

## 6  Discussion and conclusions

We have identified the complete family of Hamiltonians with a ground-state of Jastrow form, involving one and two-body functions. These models describe particles of equal mass in $d$-spatial dimensions with kinetic energy and one-, two- and three-body local potentials, that neither involve a magnetic field nor momentum-dependent interactions. For $d = 3$ this family corresponds to the Calogero-Marchioro models [27] while the corresponding family in $d = 1$ has been discussed in [16]. For arbitrary $d$ our results provide the complete family of parent Hamiltonians of Jastrow wavefunction without restriction to Calogero-like models associated with $SU(1,1)$ symmetry [34] or the nonlocal momentum dependent terms [35]. Further, while these models generally involve three-body interactions, their long-wavelength behavior is independent of the latter [35].

Our construction readily provides the generalization to arbitrary spatial dimension of known models, such as the Calogero-Sutherland, Calogero-Moser, and inverse-sinh-square models. In addition, our results greatly facilitate the identification of new specific instances within this family of models. To this end, it suffices to choose the pair function entering the Jastrow form and to evaluate its first and second derivatives. As an example, motivated by the many-body quantum bright soliton found by McGuire state in the attractive Lieb-Liniger model, we have shown that its generalization to higher dimensions has a parent Hamiltonian involving inverse distance interactions. Similarly, we have constructed novel models by considering wavefunctions functions that are the product of the corresponding ground state of some of these models. The parent Hamiltonians of the resulting models (for which we use a hybrid notation e.g., McGuire-Calogero-Sutherland, hyperbolic McGuire, etc.) have a hybrid structure with pairwise interactions inherited from the constituent models and additional cross terms. This construction can be generalized to higher-order hybrids involving more than two reference models.

Importantly, our results allow reverse-engineering the pair function that gives rise to a given pairwise potential. As an example, we have identified the ground-state of a Hamiltonian with Yukawa two-body interactions, and an additional model with a vanishing two-body term that is governed exclusively by three-body interactions.

Our results can be extended to models that are supersymmetric [47], include spin degrees of freedom, as well as multiple species [48–50], and truncated interactions [51,52]. Likewise, one can envision the extension to account for anyons with two-body interactions involving the relative angular momentum [31]. Yet another generalization is suggested by considering more general Jastrow wavefunctions of the type in Eq. (2). An exciting prospect is offered by considering Nosanov-Jastrow wavefunctions used to describe quantum solids, as this may allow the identification of quasi-exactly solvable many-body quantum systems with a lattice. To this end, one may consider including symmetrized wave-functions [16, 53–55], shadow wave-functions [56–58], and permutation-sampling methods [59, 60].

While we have focused on ground-state wavefunctions and the identification of the corresponding parent Hamiltonians, an interesting outlook concerns the identification of excited states and their corresponding energy eigenvalues. The systems discussed are generally quasi-exactly solvable and thus only part of the spectrum may be derived. It is thus of interest to explore whether one can establish the integrability of the parent Hamiltonian from the prop-

erties of the ground-state Jastrow wavefunction.

## Acknowledgements

The authors thank Aurelia Chenu, Bogdan Damski, Xi-Wen Guan, Apollonas S. Matsoukas-Roubeas, and Jing Yang for illuminating discussions. We thank P. Le Doussal for pointing out the recent reference [61] about one-dimensional fermionic ground-states and for discussing the connection of our work with his recent work [62] about diffusion of interacting particles in one dimension.

## A  Laplacian of Jastrow wavefunctions

The action of the Laplacian yields on a Jastrow wavefunction of the form $\Phi_0(r_1,\ldots,r_N) = \prod_{i<j} f(r_{ij})$ is given by

$$
\begin{aligned}
\sum_i \Delta_i \Phi_0 &= \sum_i \vec{\nabla}_i \cdot \vec{\nabla}_i \Phi_0 \\
&= \left( \sum_i \vec{\nabla}_i \cdot \sum_{j \neq i} \frac{\vec{r}_{ij}}{r_{ij}} \frac{f'_{ij}}{f_{ij}} \right) \Phi_0 + \sum_i \sum_{j \neq i} \frac{\vec{r}_{ij}}{r_{ij}} \frac{f'_{ij}}{f_{ij}} \cdot \left( \vec{\nabla}_i \Phi_0 \right) \\
&= \sum_i \sum_{j \neq i} \left( \vec{\nabla}_i \cdot \frac{\vec{r}_{ij}}{r_{ij}} \right) \frac{f'_{ij}}{f_{ij}} \Phi_0 + \sum_i \sum_{j \neq i} \frac{\vec{r}_{ij}}{r_{ij}} \cdot \vec{\nabla}_i \left( \frac{f'_{ij}}{f_{ij}} \right) \Phi_0 + \sum_i \sum_{j \neq i} \frac{\vec{r}_{ij}}{r_{ij}} \frac{f'_{ij}}{f_{ij}} \cdot \left( \vec{\nabla}_i \Phi_0 \right) \\
&= \sum_i \sum_{j \neq i} \frac{d-1}{r_{ij}} \frac{f'_{ij}}{f_{ij}} \Phi_0 + \sum_i \sum_{j \neq i} \frac{\vec{r}_{ij}}{r_{ij}} \cdot \frac{\vec{r}_{ij}}{r_{ij}} \left( \frac{f''_{ij}}{f_{ij}} - \frac{f'^2_{ij}}{f^2_{ij}} \right) \Phi_0 + \sum_i \left( \sum_{j \neq i} \frac{\vec{r}_{ij}}{r_{ij}} \frac{f'_{ij}}{f_{ij}} \right)^2 \Phi_0 \\
&= \sum_i \sum_{j \neq i} \frac{d-1}{r_{ij}} \frac{f'_{ij}}{f_{ij}} \Phi_0 + \sum_i \sum_{j \neq i} \left( \frac{f''_{ij}}{f_{ij}} - \frac{f'^2_{ij}}{f^2_{ij}} \right) \Phi_0 \\
&\quad + \sum_i \sum_{j \neq i} \left( \frac{\vec{r}_{ij}}{r_{ij}} \frac{f'_{ij}}{f_{ij}} \right)^2 \Phi_0 + \sum_i \sum_{j \neq k \neq i} \frac{\vec{r}_{ij}}{r_{ij}} \cdot \frac{\vec{r}_{ik}}{r_{ik}} \frac{f'_{ij}}{f_{ij}} \frac{f'_{ik}}{f_{ik}} \Phi_0 \\
&= \sum_i \sum_{j \neq i} \left( \frac{f''_{ij}}{f_{ij}} + \frac{d-1}{r_{ij}} \frac{f'_{ij}}{f_{ij}} \right) \Phi_0 + \sum_i \sum_{j \neq k \neq i} \frac{\vec{r}_{ij}}{r_{ij}} \cdot \frac{\vec{r}_{ik}}{r_{ik}} \frac{f'_{ij}}{f_{ij}} \frac{f'_{ik}}{f_{ik}} \Phi_0 \; .
\end{aligned}
$$

(90)

(91)

A similar derivation holds for the generalized Jastrow wavefunction $\Psi_0 = \prod_{i<j} f(r_{ij}) \prod_k g(r_k) = \prod_k g(r_k) \Phi_0$. We first evaluate the gradient:

$$
\vec{\nabla}_i \Psi_0 = \sum_{j \neq i} \left( \frac{\vec{\nabla}_i f_{ij}}{f_{ij}} \right) \Psi_0 + \left( \frac{\vec{\nabla}_i g_i}{g_i} \right) \Psi_0 = \left( \sum_{j \neq i} \frac{\vec{r}_{ij}}{r_{ij}} \frac{f'_{ij}}{f_{ij}} + \frac{\vec{r}_i}{r_i} \frac{g'_i}{g_i} \right) \Psi_0 \; .
$$

Using this expression, the Laplacian is found to be

$$
\begin{aligned}
\Delta_i \Psi_0 &= \vec{\nabla}_i \cdot \vec{\nabla}_i \Psi_0 \\
&= \left[ \sum_{j \neq i} \vec{\nabla}_i \left( \frac{\vec{r}_{ij}}{r_{ij}} \frac{f'_{ij}}{f_{ij}} \right) + \vec{\nabla}_i \left( \frac{\vec{r}_i}{r_i} \frac{g'_i}{g_i} \right) \right] \Psi_0 + \left( \sum_{j \neq i} \frac{\vec{r}_{ij}}{r_{ij}} \frac{f'_{ij}}{f_{ij}} + \frac{\vec{r}_i}{r_i} \frac{g'_i}{g_i} \right)^2 \Psi_0
\end{aligned}
$$

$$
\begin{aligned}
= \quad & \sum_{j\neq i}\left[\frac{d-1}{r_{ij}}\frac{f'_{ij}}{f_{ij}}+\frac{f''_{ij}}{f_{ij}}-\frac{f'^{2}_{ij}}{f^{2}_{ij}}\right]\Psi_0+\left[\frac{d-1}{r_i}\frac{g'_i}{g_i}+\frac{g''_i}{g_i}-\frac{g'^{2}_i}{g^{2}_i}\right]\Psi_0 \\
& +\sum_{j\neq k\neq i}\left(\frac{\vec{r}_{ij}}{r_{ij}}\cdot\frac{\vec{r}_{ik}}{r_{ik}}\frac{f'_{ij}}{f_{ij}}\frac{f'_{ik}}{f_{ik}}\right)\Psi_0+2\sum_{j\neq i}\left(\frac{\vec{r}_{ij}}{r_{ij}}\frac{f'_{ij}}{f_{ij}}\cdot\frac{\vec{r}_i}{r_i}\frac{g'_i}{g_i}\right)\Psi_0+\sum_{j\neq i}\frac{f'^{2}_{ij}}{f^{2}_{ij}}\,\Psi^2_0+\frac{g'^{2}_i}{g^{2}_i}\,\Psi_0 \\
= \quad & \sum_{j\neq i}\left[\frac{d-1}{r_{ij}}\frac{f'_{ij}}{f_{ij}}+\frac{f''_{ij}}{f_{ij}}\right]\Psi_0+\sum_{j\neq k\neq i}\left(\frac{\vec{r}_{ij}}{r_{ij}}\cdot\frac{\vec{r}_{ik}}{r_{ik}}\frac{f'_{ij}}{f_{ij}}\frac{f'_{ik}}{f_{ik}}\right)\Psi_0 \\
& +2\sum_{j\neq i}\left(\frac{\vec{r}_{ij}}{r_{ij}}\frac{f'_{ij}}{f_{ij}}\cdot\frac{\vec{r}_i}{r_i}\frac{g'_i}{g_i}\right)\Psi_0+\left[\frac{d-1}{r_i}\frac{g'_i}{g_i}+\frac{g''_i}{g_i}\right]\Psi_0\ .
\end{aligned}
\tag{92}
$$

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
