# Peer review of "Parent Hamiltonians of Jastrow Wavefunctions"

_SciPost Physics, doi:SciPost Phys. Core 4, 030 (2021)_

## Round 1 · Referee Report · Anonymous (Referee 1) · 2021-8-16

Strengths

The authors of this manuscript presented a systematic description of the recent well-established method to construct a complete family of many-body quantum Hamiltonians with ground-state of Jastrow form involving the pairwise product of a pair function in an arbitrary spatial dimension. In section 2, the parent Hamiltonians with 2-body & 3-body pairwise potentials in d-spatial dimensions are constructed. In section 3, the 1-body term serving as external potential in the Hamiltonian is introduced by adding one particle term to the Jastrow form of the wave function, Consequently, the long-range contributions are involved by mixing the 2- and 1-body couplings in the Hamiltonians. Some simple examples were mentioned in this section. The section 4 was devoted to construct 9 models by using the method discussed in previous sections, among which the first 4 have been well studied in literatures, for example, see “PHYSICAL REVIEW RESEARCH 2, 043114 (2020)” etc., and the last 5 ones are newly constructed. However, these are in fact not totally new. They can be a kind of combinations of previous 4 models. In the section 5, they illustrated how to construct the explicit Jastrow form wave function once the interaction is known.

Weaknesses

The weakest part of this paper is their method. It is not a new method and such kind of Hamiltonians have been studied before. In addition, the physical meaning of these models was not explained.

Report

In view of the systematic construction of such kind of systems, I see that the manuscript was well organized and written. The key merit of this paper is that the parent Hamiltonian construction was systematically generalized to any spatial dimensions. In the 1d case, the newly constructed Hamiltonians involve 2-body interaction (Eq.11) and external potential (Eq.17). while the 2-body interaction and external potential are seen in higher dimensions (d>1) case.
I shall be happy to recommend their revised version of this submission for publication in SciPost.

Requested changes

At least, the authors should discuss physical understanding and a possible application of such constructed Hamiltonians in more details. They should also anticipate how such kind of wave functions can be used to calculate the correlation functions.

  • validity: high
  • significance: good
  • originality: good
  • clarity: high
  • formatting: good
  • grammar: excellent

Author:  Mathieu Beau  on 2021-09-23  [id 1775]

(in reply to Report 1 on 2021-08-16)

See attached pdf

Attachment:

Ref1.pdf

---

## Round 1 · Referee Report · Anonymous (Referee 2) · 2021-8-31

Strengths

1) In this manuscript, the authors find the family of many-body Hamiltonians with ground-state of Jastrow form in arbitrary spatial dimensions. This extends previous results known for one-dimensional and for three-dimensional systems only, extending the cases considered in Journal of Mathematical Physics 16(5), 1172 (1975) and in Phys. Rev. B 43, 3255 (1991). 2) The authors also show that, in some cases, the pair-function can be reversed engineered, finding a wave-function corresponding to a given two-body potential.

Weaknesses

1) Most results reported in this manuscript represent a generalization of previously known results.

Report

The authors find the family of many-body Hamiltonians with ground-state of Jastrow form in arbitrary spatial dimensions, finding that, in general, these Hamiltonians include two and three body interactions. This finding extends the Calogero-Marchioro result, which was valid only for 3D, to arbitrary dimensions. The authors also discuss the case of wave-function with one-particle terms, leading to long-range interactions in the parent Hamiltonian. Several models are discussed. Interestingly, they discuss the reverse-engineering of the Jastrow pair function corresponding to a parent Hamiltonian.

The findings presented in this manuscript are interesting and sound. However, it is fair to say that they represent a generalization of previously know results.
I find that the manuscript is definitely suitable for publication in SciPost Physics Core, provided that the comments reported in the "Requested Changes" are adequately addressed in a revised version. In order for me to give a strong recommendation for publication in the flagship journal SciPost Physics, the authors should better emphasize the relevance of the generalized results, and discuss more in depth the physics of at least one of the novel models introduced in this manuscript. Can the authors provide some interesting predictions for these models? Such predictions would highlight the relevance of the techniques discussed in the manuscript.

Requested changes

1) In the introduction, the authors state that the Jastrow wave-function with only two-body terms is suitable to describe quantum solids. It is my understanding that, in fact, this wave-function only captures the properties of fluid states. As mentioned by the authors in the conclusions, the Nosanov-Jastrow wave-function is instead suitable to describe the solid state.

2) In the introduction, the authors write "Slater determinants of such Jastrow functions are also widely used in and quantum chemistry." First, there is perhaps a typo ("...in and..."). More importantly, this statement is not clear. In fact, electronic systems are often described via products of Jastrow functions and Slater determinants of single-particle wave-functions. Fermionic (i.e., antisymmetric) wave-functions can also be built starting from pair orbitals, but using, in general, Pfaffian wave-functions (PRL 96, 130201 (2006), J. Chem. Theory Comput. 16.10, 6114-6131 (2020)).

3) In the conclusion, the authors mention the Nosanov-Jastrow wave-function for bosonic solid states. However, it is worth mentioning that the original model does not satisfy the bosonic symmetry. In order to account for Bose-Einstein statistics, various approaches have been introduced, including symmetrized wave-functions (J. Stat. Mech. P07003 (2005), NJP 11 013047 (2009)), shadow wave-functions (PRB 38, 4516 (1988), PRL 60, 1970 (1988), PRB 71 140506 (2005)) and permutation-sampling methods (PRB 17 1070 (1978), PRL 108, 155301 (2012)).

4) The authors use the term quasi-exactly solvable models. The meaning is elucidated only at the end of the manuscript, and it refers to models for which only part of the spectrum is obtained. This definition should be given earlier. Also, it is not clear if, in fact, in most cases only the ground-state energy is known, and if its evaluation requires additional computations (e.g., Monte Carlo sampling of the Jastrow wave-function.)

  • validity: high
  • significance: good
  • originality: good
  • clarity: high
  • formatting: excellent
  • grammar: excellent

Author:  Mathieu Beau  on 2021-09-23  [id 1776]

(in reply to Report 2 on 2021-08-31)

See attached pdf

Attachment:

Ref2.pdf

---

## Round 2 · Author Response

We thank the referees for the accurate summary of our contribution and for useful suggestions.

The novel models we have introduced have no precedence in the literature to the best of our knowledge. We term the models according to names that allows to quickly appreciate the relation with some other models, but the additive character of the interactions cannot be taken for granted and generally mixing terms occur.
Nonetheless, the strength of our contribution is in facilitating the systematic construction of new models: choose your favorite one- and and two-body functions in the Jastrow form and find the parent Hamiltonian through the equations we derive.
For historical accidents, the generality of the formalism and the results we report have not been presented in spite of the various related attempts documented in the introduction starting in 1975 and sustained to this day. In this sense, we close this effort by providing the complete family, which is infinite. Our examples are chosen to show how instances of known models are included in our formalism and how to generate new models in these infinite family.

In addition, we wish to draw the attention of the referee to the work by Kane et al. which established that parent Hamiltonians of Jastrow wavefunctions share the same long-wavelength behavior than that of the same Hamiltonian in the absence of three-body interactions. That makes the resulting models physically appealing: the physics is set by the two-body interactions. Further, the family is infinite ans instances can be found by reverse engineering for given interactions. In view of these results and the long-time impact of the preceding related works on the topic, it seems clear that our work is an important contribution meriting publication in SciPost Physics. In some sense, it closes a search started in 1975 by Calogero by identifying the complete family of parent Hamiltonians of Jastrow wavefunctions in any spatial dimension.
To put in perspective our results, it is important to notice that the more limited Calogero construction is actually a reference result and in a sense a textbook result (see ref. [13] and [29] among many others). In addition, as we mention in the discussion, our work has a plethora of ramifications.

---

## Round 2 · List of Changes

Introduction:
- we expanded the discussion on Jastrow functions and added two references, see [11] and [12].
- we added a sentence on the quasi-solvability of models given by equation (1).

Conclusion:
- we have expanded the discussion in this revision to reflect various approaches in quantum solids. We added seven new references [55-61].
- We emphasized that the system are quasi-exactly solvable and added the following sentence: "The systems discussed are generally quasi-exactly solvable and thus only part of the spectrum may be derived"

Acknowledgment:
- We added: "We thank P. Le Doussal for pointing out the recent reference [61] about one-dimensional fermionic ground-states and for discussing the connection of our work with his recent work [62] about diffusion of interacting particles in one dimension."

---

## Editorial Decision

published